# Explaining public dental service utilization: A theoretical model

**Maria Helena Rodrigues Galvão** [1,2]*, **Angelo Giuseppe Roncalli**[1]

**1** Postgraduate Program in Public Health, Federal University of Rio Grande do Norte, Natal, Rio Grande do Norte, Brazil, **2** Academic Center of Vitória, Federal University of Pernambuco, Vitória de Santo Antão, Pernambuco, Brazil

\* mhrgalvao@gmail.com

## Abstract

### Objectives

Constructing and validating a theoretical model of relationships between dental services use and socioeconomic characteristics, oral health status, primary care coverage, and public dental services.

### Methods

The first stage of the study consisted of developing a theoretical-conceptual model to demonstrate the expected relationships between variables based on the literature. In the second stage, we tested the proposed theoretical model using the Partial Least Squares Structural Equation Modeling (PLS-SEM) technique, using data from the Brazilian National Health Survey conducted in 2019 with a sample of 41,664 individuals aged 15 or older.

### Results

This study successfully defined a theoretical model that explains the systematic relationships involving public dental services utilization. Socioeconomic status was negatively associated with oral health status ($β = -0.376$), enrollment in primary care facilities ($β = -0.254$), and the use of public dental consultations ($β = -0.251$). Being black, indigenous, or living in a rural area was directly associated with lower socioeconomic status and greater use of public dental services.

### Conclusions

The identified relationships, establishing a theoretical basis for further investigations, also provide evidence of a public access policy's effect on oral health services on equity, supporting the construction of more effective and equitable public policies.

## Introduction

Oral diseases are the most prevalent diseases in the world [1]. Considering only untreated dental caries, approximately 2.3 billion people require dental care worldwide [2]. Furthermore,

**Data Availability Statement:** Research data are publicly available on the Brazilian Institute of Geography and Statistics website (www.ibge.gov. br).

**Funding:** YES. This study was funded in part by the Coordenação de Aperfeiçoamento de Pessoal de

Nível Superior—Brazil (CAPES; finance code 001), with a postgraduate fellowship and payment of publication fees. This funding did not interfere with study design, data collection, analysis, interpretation, and manuscript writing.

**Competing interests:** The authors have declared that no competing interests exist.

economically vulnerable groups experience a disproportionate burden of oral diseases [2]. They are also observed to utilize dental services less frequently [3], highlighting the presence of a social gradient in oral health.

The utilization of dental services has been studied for a long time. Since the 1980s, models have been developed to explain this phenomenon [4]. Behavioral healthcare utilization models provide a theoretical framework for defining the variables evaluated in studies with healthcare utilization as an outcome [5]. These models allow for the specification of relationships between the analyzed variables, facilitating the evaluation of programs and policies related to healthcare service access and utilization [5].

Developed in the 1960s, the Andersen Behavioral Model is the primary behavioral model for healthcare service utilization. The fundamental premise of the Andersen Model posits that people's use of services is a function of their predisposition to use services (predisposing factors), factors that enable or impede the use (enabling factors), and their need for care (need factors) [5]. In Andersen's classic behavioral model, dental service utilization is considered discretionary and less important than medical services. It is explained by social conditions, beliefs, financial resources, and symptom severity [6].

As a reflection of the low political priority for global oral health, dental care is excluded in most healthcare systems and available health insurance models [7]. In countries that have included dental care in their healthcare systems, there is still a requirement for co-payments, restricted service baskets, or coverage only for specific and vulnerable groups [8].

The coverage of dental services is a recent topic, where there is advocacy for these services to be included in the essential service baskets of the proposed universal health coverage [8, 9]. It is essential to understand the effectiveness of this protection model as a public policy, especially in the context of universal healthcare systems. Evidence production is needed to point to the effectiveness of implementing such services, given the high cost and health needs of the population.

There are gaps in knowledge regarding theoretical models that explain the utilization of public dental services. Several studies apply Andersen's behavioral model [10], which, although it has its contribution and application in different areas of knowledge, does not clearly explain the phenomenon of public services utilization. Furthermore, it was developed and updated with United States data, which does not have a public and universal health system [11]. If, on the one hand, dental services' utilization is generally marked by great inequalities, opposite relationships are observed when evaluating the utilization of public and free health services [12].

This study aimed to construct and validate a conceptual framework to evaluate the utilization of public dental services. Moreover, to better comprehend the interdependent relationships between social and socioeconomic characteristics, oral health status, primary care coverage, and public dental services by Brazilian adolescents, adults, and older adults.

## Method

The research consists of an analytical study, proposing a theoretical model and its statistical validation through structural equation modeling. The first stage of the study consisted of developing a theoretical-conceptual model to demonstrate the expected relationships between the variables. It was specified which variables are related or unrelated to the construction of the measurement and structural models of the validation process. Scientific evidence published in articles on the subject and the researchers' previous experience were used to construct the model. The relationships already consolidated in the literature on the use of dental services and considered in the proposition of this model were education [10, 13–22], income [10, 13,

15, 17–19, 22, 23], area of residence (rural or urban) [13, 15, 16, 18, 19, 21, 24], sex [10, 13, 15–21], skin color [13–18, 21, 22], enrollment in the primary care facilities or coverage of public services [13, 18, 21, 22], edentulism or the number of lost teeth [6, 17, 19, 22, 25, 26] and self-assessment of oral health [13, 18, 19, 24].

The study's main outcome was using public dental services through the Brazilian National Health System. This outcome was measured by a construct formed by the variable "Was your last dental consultation carried out through the Public Health System?" and the associated measurement error.

The socioeconomic position, a mediator construct, consists of a reflective construct composed of two indicator variables "years of study" and "dollars per capita income". The oral health condition is a mediating reflective construct with two indicator variables "number of lost teeth" and "oral health self-assessment". The individual variables of skin color, area of residence, and sex were assessed by a construct formed by the corresponding indicator variable and the associated measurement error. The mediator construct "household enrolled in primary care" was formed by the corresponding indicator variable and the associated measurement error.

These variables are present in the National Health Survey (PNS) database, which was used as the basis for the model validation stage. The PNS consists of a nationally representative household survey conducted every five years in Brazil since 2013, with the last edition in 2019. The survey uses a three-stage cluster sampling, with the first stage being the Primary Sampling Unit (PSU), the census tracts, the second stage being households, and the third stage being a selected resident aged 15 years or older [27]. The total sample used in this study was 41,664 individuals selected to answer the survey questionnaire and reported having had a dental consultation in the past 12 months. The data used in the study do not identify the participants, are in the public domain, and are available on the Brazilian Institute of Geography and Statistics (IBGE) website. Table 1 presents the variables included in the theoretical model, their description, and the categorizations adopted.

The Partial Least Squares Structural Equation Modeling (PLS-SEM) technique was used to test the proposed theoretical model. This technique was chosen because it is a non-parametric method that does not assume normality distribution for the variables included in the model since it includes quantitative, ordinal, and binary categorical variables. PLS-SEM does not require the model identification requirements necessary for the conventional Covariance-Based Structural Equation Modeling (CB-SEM) technique. This allows estimating constructs with a single indicator variable, as in this study. PLS-SEM is also considered the most appropriate technique for working with studies using secondary data [28].

The measurement model was estimated to describe the relationships between the latent constructs and the indicator variables. The structural model was estimated to describe the constructs' relationships (or paths). The validity and reliability assessment of the measurement model was based on standard measures. Since PLS-SEM is a non-parametric method, bootstrapping was performed to estimate the standard errors and the parameter's confidence intervals. The parameters of the final model were assessed after 42,000 bootstrap sub-samples. As PLS-SEM processes standardized data, the path coefficients indicate the changes in the values of an endogenous construct associated with changes in one standard deviation unit of a given predictor construct while keeping all other predictor constructs constant. Statistical analyses were performed using the free software R version 4.2.0, with the "SEMinR" package.

## Results

Table 2 presents the sample characterization of the 41,664 individuals included in the study.

**Table 1. Description of the study's variables and adaptation strategies for the analysis model.** Brazil, 2019.

| Variable | Description | Categorization |
|---|---|---|
| Sex | Sex | 0-Female<br>1-Male |
| Skin Color/Race Black | Self-reported skin color | 0-White, Asian, Indigenous<br>1-Black (Black or Brown) |
| Skin Color/ Race Indigenous | Self-reported skin color | 0-White, Yellow, Black, or Brown<br>1-Indigenous |
| Years of study | Years of study completed | 0 to 16 years |
| Family income *per capita* | *Per capita* household income, converted in dollars considering purchasing power parity in December/2019 | Continuous variable |
| Area of residence | Area of residence | 0-Urban<br>1-Rural |
| Enrolled in Primary Care Facilities | Household enrollment in Primary Care Facilities | 0-Yes<br>1-No/Do not know |
| Self-rated oral health | Self-assessment of oral health | 0-Very good<br>1-Good<br>2-Moderate<br>3-Bad<br>4-Very bad |
| Number of missing teeth | Number of self-reported missing teeth | Discrete variable<br>0–32 teeth |
| Public Dental Services utilization | The last dental consultation was made by SUS (the Portuguese acronym for the Brazilian National Health System) | 0-No<br>1-Yes |

The PLS-SEM algorithm performed nine iterations until the proposed model's convergence, representing a stable solution. The reflective latent construct "oral health conditions" showed suitability to reliability and validity criteria. When evaluating the reliability of the indicator variables for this construct, we observed that the loads for the "self-assessment of oral health" (0.737) and "tooth loss" (0.814) indicated sufficient levels of reliability. Regarding the reliability and internal consistency of the construct, we found good values (rhoc = 0.752). The average variance extracted for the indicator was 0.603, above the cutoff point of 0.5, indicating that the construct explains 60.3% of the variance of the variables that compose it, which represents satisfactory convergent validity.

The latent construct "Socioeconomic Position" also showed satisfactory reliability and validity criteria values. The factorial loads for the indicator variables "years of study" (0.686) and "per capita income" (0.901) indicate sufficient levels of reliability, considering the proposed model. Regarding the reliability and internal consistency of the construct, we found good values (rhoc = 0.778). The average variance extracted for the indicator was 0.641, above the cutoff point of 0.5, indicating that the construct explains 64.1% of the variance of the variables that compose it, which represents satisfactory convergent validity. Latent constructs formed by a single indicator variable have reliability values of 1.0. All constructs presented discriminant validity, with heterotrait-monotrait ratio of correlations (HTMT) values lower than 0.85.

Table 3 presents the estimated path coefficients by the model between the exogenous and endogenous latent model constructs. When evaluating the t-statistic values and the confidence intervals, we observed that all estimated path parameters are statistically significant. The socioeconomic position was associated with an oral health condition, presenting the highest observed effect (β = -0.376). The worse the socioeconomic position, represented by fewer years

**Table 2. Descriptive analysis of the variables included in the study.** Brazil, 2019.

| Variable | n<sup>not weighted</sup> | n<sup>weighted</sup> | % | | CI 95% |
|---|---|---|---|---|---|
| Use of public dental services | | | | | |
| Yes | 11.161 | 19.264.898 | 23,1 | | (22,3; 23,9) |
| No | 30.435 | 64.103.920 | 76,9 | | (76,1; 77,7) |
| Sex | | | | | |
| Female | 23.904 | 47.293.849 | 56,6 | | (55,7–57,4) |
| Male | 17.765 | 36.330.798 | 43,4 | | (42,6–44,3) |
| Black Color/Race | | | | | |
| Yes | 23.946 | 42.559.345 | 50,9 | | (49,8–52,0) |
| No | 17.723 | 41.065.302 | 49,1 | | (48,0–50,2) |
| Indigenous Color/Race | | | | | |
| Yes | 294 | 352.441 | 0,4 | | (0,3–0,5) |
| No | 41.375 | 83.272.206 | 99,6 | | (99,5–99,7) |
| Area of residence | | | | | |
| Urban | 34.413 | 74.691.526 | 89,3 | | (88,8–89,8) |
| Rural | 7.256 | 8.933.121 | 10,7 | | (10,2–11,2) |
| Enrollment in Primary Care Facilities | | | | | |
| Yes | 25.199 | 49.284.592 | 58,9 | | (57,6–60,6) |
| No / Do not know | 16.470 | 34.340.055 | 41,1 | | (39,7–42,4) |
| Self-rated oral health | | | | | |
| Very good | 6105 | 14.049.701 | 16,8 | | (16,1–17,6) |
| Good | 24.731 | 49.238.201 | 58,9 | | (58,0–59,8) |
| Regular | 9314 | 17.721.810 | 21,2 | | (20,5–21,9) |
| Bad | 1289 | 2.252.768 | 2,7 | | (2,4–3,0) |
| Very bad | 230 | 362.166 | 0,4 | | (0,4–0,5) |
| | | | Average | SD | CI 95% |
| Number of missing teeth | 41.669 | 83.624.648 | 4,74 | 0,068 | (4,65–4,92) |
| Family income *per capita* | 41.669 | 83.624.648 | 864,69 | 16,949 | (831,46–897,91) |
| Years of study | 41.669 | 83.624.648 | 11,06 | 0,042 | (10,97–11,14) |

**Table 3. Estimates of path coefficient, significance, and confidence intervals.**

| Paths | Model estimation | *Bootstrap* Estimates | | | |
|---|---|---|---|---|---|
| | Path coefficient | Average | Standard deviation | Statistic "t" | CI 95% |
| Socioeconomic position to Oral Health Condition | -0,376 | -0,376 | 0,005 | -78,344 | -0,385; -0,366 |
| Rural Area to Socioeconomic Position | -0,270 | -0,270 | 0,004 | -64,321 | -0,279; -0,262 |
| Socioeconomic Position to Enrolment in primary health care | -0,254 | -0,255 | 0,005 | -53,287 | -0,264; -0,245 |
| Socioeconomic Position to public dental services utilization | -0,251 | -0,251 | 0,005 | -51,857 | -0,260; -0,242 |
| Black Skin Color to Socioeconomic Position | -0,194 | -0,194 | 0,005 | -35,342 | -0,205; -0,183 |
| Rural area to public dental services utilization | 0,144 | 0,144 | 0,005 | 26,163 | 0,133; 0,155 |
| Enrolment in primary health care to Public dental services utilization | 0,117 | 0,117 | 0,005 | 25,950 | 0,108; 0,126 |
| Black Skin Color to Public dental services utilization | 0,072 | 0,072 | 0,005 | 15,543 | 0,063; 0,081 |
| Male Sex to Public dental services utilization | -0,045 | -0,045 | 0,004 | -10,025 | -0,053; -0,036 |
| Indigenous race/skin color to Socioeconomic Position | -0,038 | -0,038 | 0,004 | -9,010 | -0,046; -0,030 |
| Oral Health Condition to Public dental services utilization | 0,036 | 0,036 | 0,005 | 6,889 | 0,026; 0,046 |
| Indigenous race/skin color to Public dental services utilization | 0,035 | 0,035 | 0,005 | 6,902 | 0,025; 0,045 |

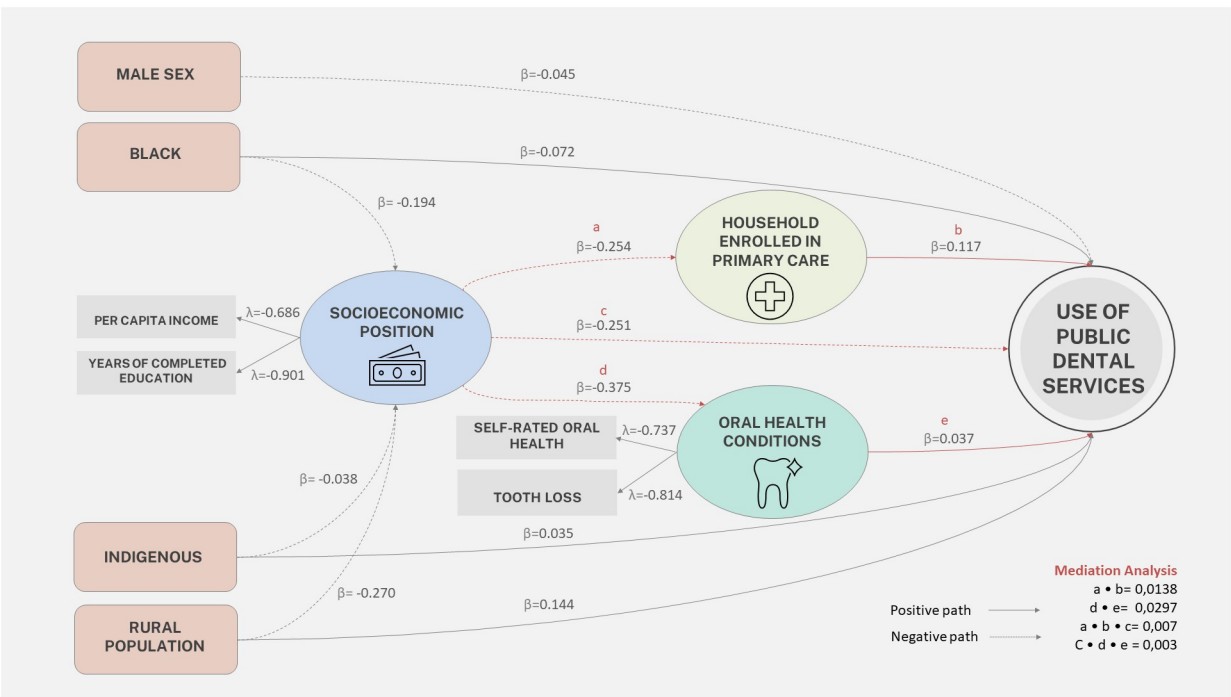

**Fig 1. Pathways diagram that explains the utilization of public dental services.**

of study and lower per capita income, the worse the oral health condition. The socioeconomic position was also associated with enrollment in primary care facilities by people with lower socioeconomic positions (β = -0.254). The lower socioeconomic position was also associated with dental consultations in the public health system (β = -0.251). Being black, indigenous, or residing in a rural area was directly associated with a lower socioeconomic position and the use of public dental services.

Fig 1 presents the final diagram of the pathways that explain the utilization of public dental services in Brazil, and Table 4 shows its decomposition into direct, indirect, and total effects. Socioeconomic position has a negative effect on the use of public dental services (β = -0.294). This effect is 85.7% direct and 14.3% indirect, mediated by the oral health condition (β = -0.013) and enrollment in the primary care facilities (β = -0.029). Both direct and indirect effects were significant, indicating partial mediation. Enrollment in primary care facilities and

**Table 4. Indirect and total effects of latent constructs in public dental services utilization.**

| Constructs | Direct effects | | Indirect effects | | Total effects (IC95%) |
|---|---|---|---|---|---|
| | | % | | % | |
| Black Skin Color | 0,072 | 55,8 | 0,057 | 44,2 | 0,129 (0,120; 0,138) |
| Rural area | 0,144 | 64,6 | 0,080 | 36,4 | 0, 223 (0,213; 0,234) |
| Indigenous race/skin color | 0,035 | 76,1 | 0,011 | 23,9 | 0,046 (0,036; 0,057) |
| Socioeconomic Position | -0,251 | 85,4 | -0,044 | 14,6 | -0,294 (-0,303;-0,286) |
| Male sex | -0,045 | 100% | 0 | - | -0,045 (-0,053; -0,036) |
| Oral Health Condition | 0,037 | 100% | 0 | - | 0,037 (0,026; 0,047) |
| Enrolment in primary health care | 0,117 | 100% | 0 | - | 0,117 (0,108; 0,126) |

oral health conditions partially mediators the effect of socioeconomic position on public dental services utilization. Both direct and indirect effects act in the same direction as the main effect, where the lower socioeconomic position is associated with higher utilization of dental services, even considering the effect of mediators. A competitive partial mediation of enrollment in primary care facilities was observed in the relationship between socioeconomic position and utilization of dental services. A complementary partial mediation effect of the oral health condition in the relationship between socioeconomic position and utilization of dental services.

When evaluating the explanatory model power based on the $R^2$ of the endogenous constructs, we found values higher than 0.10 for the utilization of public dental services constructs ($R^2 = 0.168$), socioeconomic position (R2 = 0.120), and oral health condition (R2 = 0.141). The predictive power was low for the primary care facility enrollment construct ($R^2 = 0.065$). However, we emphasize that the current model has no intention of predicting this outcome, only evaluating its role as a mediator of other relationships (28,29). All Variance Inflation Factor (VIF) values found for the structural model were less than 3, demonstrating that collinearity between predictor constructs is not a critical issue for the model.

## Discussion

The present study successfully defined a theoretical model that explains the relationships between the studied variables and the use of public dental services, highlighting the mediating role of socioeconomic status as a central element in this theory.

The lower socioeconomic position had a direct main effect on the utilization of public dental services. However, enrollment in primary health care has the potential to mitigate this effect through partial complementary mediation, indicating that in the presence of such enrollment, individuals with lower socioeconomic positions utilize public dental services more frequently. Oral health condition also works as a partial complementary mediator, as individuals with lower socioeconomic position utilize more public dental services in the presence of poorer oral health conditions. The socioeconomic position also mediates the relationships between sociodemographic variables (race, skin color, and area of residence) and the utilization of public dental services.

In England, Wales, and Northern Ireland, a predominant direct effect of socioeconomic position on oral health was also observed, with lower socioeconomic position associated with worse self-rated oral health and fewer natural teeth. This reinforces the evidence of this relationship in other populations [29]. Although the main objective of this study was not to assess the use of dental services, a significant direct pathway was observed between dental assistance and socioeconomic position, indicating that adults with lower socioeconomic position visit the dentist less frequently and are less likely to have regular check-ups, despite having worse self-rated oral health [29].

In our study, however, considering the utilization of public dental services, we found the same effect in the opposite direction, where adults with lower socioeconomic position and worse oral health conditions use public dental services more often. A previous study [12] has presented evidence suggesting that public dental services might help to reduce disparities in accessing dental appointments. The study [12] found that people with lower levels of education, lower family income, indigenous, black or brown people, living in rural areas, with poorer oral health, enrolled in primary health care, and living in Federation states with a higher percentage of the population covered by oral health teams in primary care are more likely to use public dental services. Considering the utilization of public and private dental services, the association is reversed [13–15]. Furthermore, the relationship between these variables has not been previously presented in other explanatory public dental services utilization models.

The measurement model used in the study demonstrated that the proposed constructs of "oral health condition" and "socioeconomic position" were accurately represented by the indicator variables, as shown through convergent, discriminant, and nomological validity. This has two practical implications: firstly, it enables understanding the theoretical relationship between the number of lost teeth and self-rated oral health, as well as household per capita income and education, expressed by the observed covariation. Secondly, this finding supports using these constructs in future studies, particularly those that rely on health survey secondary data.

Similarly, the structural model demonstrated that the relationship pattern of the variables included in the theoretical model fits the reality considering the analyzed sample, which is comprehensive and representative of the Brazilian population, demonstrating statistical evidence of validity for the proposed model. This result has two key implications. Firstly, it empirically confirms the proposed theoretical relationships, which enables a better understanding of the use of public dental services and provides a theoretical basis for future investigations on that subject. Secondly, it allows for evaluating the proposed associations' strength, direction, and effects, providing new evidence for explaining the phenomenon under study.

As an original finding, we demonstrated that, when evaluating the use of public services, inverse effects were observed compared to those observed in the general use of dental services, except for lower use by men. Although the association between worse socioeconomic status and worse oral health is well-established in the literature [29–31], this study innovates by demonstrating that socioeconomic status was negatively associated with enrollment in primary care and the use of public services. These findings highlight that providing public dental services in a universal system, such as the Brazilian one, is a protective policy promoting equity. This policy is more accessible to individuals with lower income. It effectively addresses complex interactions that contribute to inequalities in oral health by reaching those with greater needs and lower income, thereby providing evidence of its effectiveness.

As limitation of this study, it is that it used secondary data, which restricted the variables included in the model to those available in the PNS database. Furthermore, the variables used in the study were mostly self-reported and may have been influenced by information and memory bias, especially for the variable "number of missing teeth". However, these biases were expected to be randomly distributed, which minimized their impact on an association study. It should also be acknowledged that although Structural Equation Modeling is often considered a causal modeling technique, establishing causal relationships is only possible when specific requirements, such as covariation, non-spurious correlation, theoretical support, and temporal sequence, are met [32]. The cross-sectional design of this study, where outcomes and exposures were evaluated simultaneously, did not allow for the establishment of causal relationships.

As a strength of this study, it should be highlighted that the sample is comprehensive and nationally representative in a country with a public and free-of-charge healthcare system that offers universal coverage of dental services, where significant inequalities might be better evaluated. Additionally, the data analysis technique allowed for modeling the model complexity by simultaneously estimating multiple and interrelated dependence relationships, representing unobserved concepts, explaining measurement errors in the estimation process, and defining a theoretical model to explain the entire set of relationships [32].

From the perspective of knowledge translation, it should be **noted** that although the Brazilian healthcare system has a universal and egalitarian nature, it faces persistent challenges regarding inequality, not only in the socioeconomic context but also in access to health services. Evidence on the effect of a public access policy to oral health services on equity (a core principle of the Brazilian National Health System), highlights the necessity to include this

component in any study concerning associated factors with healthcare utilization. It also emphasizes the importance of developing an appropriate theoretical model and statistical approach that identifies associated factors and explains the relationships between them, their strength, and their latent constructs. This will enable more significant subsidies to construct more effective and equitable public policies.

## Acknowledgments

Thanks to the developers of the SEMinR Package, Dr. Soumya Ray and Dr. Nicholas P. Danks, for support in the data analysis step, correcting flaws that allowed the model to be specified. Thanks to professors Dr. Isabelle Ribeiro, Dr. Roger Keller Celeste, Dr. Kenio Costa Lima, Dr. Hélder Henrique Costa Pinheiro, Dr. Sônia Cristina Lima Chaves and Dr. Fernando José Herkrath for their contributions to the evaluation process of this work.

## Author Contributions

**Conceptualization:** Maria Helena Rodrigues Galvão.

**Formal analysis:** Maria Helena Rodrigues Galvão.

**Investigation:** Maria Helena Rodrigues Galvão.

**Supervision:** Angelo Giuseppe Roncalli.

**Writing – original draft:** Maria Helena Rodrigues Galvão.

**Writing – review & editing:** Angelo Giuseppe Roncalli.

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
