## [Decision Letter · Decision Letter 0]

15 Aug 2023

PONE-D-23-19396EXPLAINING PUBLIC DENTAL SERVICE UTILIZATION: A THEORETICAL MODELPLOS ONE

Dear Dr. Galvao,

Thank you for submitting your manuscript to PLOS ONE. After careful consideration, we feel that it has merit but does not fully meet PLOS ONE’s publication criteria as it currently stands. Therefore, we invite you to submit a revised version of the manuscript that addresses the points raised during the review process. Please submit your revised manuscript by Sep 29 2023 11:59PM. If you will need more time than this to complete your revisions, please reply to this message or contact the journal office at plosone@plos.org. Please include the following items when submitting your revised manuscript:A rebuttal letter that responds to each point raised by the academic editor and reviewer(s). You should upload this letter as a separate file labeled 'Response to Reviewers'.A marked-up copy of your manuscript that highlights changes made to the original version. You should upload this as a separate file labeled 'Revised Manuscript with Track Changes'.An unmarked version of your revised paper without tracked changes. You should upload this as a separate file labeled 'Manuscript'.If applicable, we recommend that you deposit your laboratory protocols in protocols.io to enhance the reproducibility of your results. Protocols.io assigns your protocol its own identifier (DOI) so that it can be cited independently in the future. For instructions see: https://journals.plos.org/plosone/s/submission-guidelines#loc-laboratory-protocols. Additionally, PLOS ONE offers an option for publishing peer-reviewed Lab Protocol articles, which describe protocols hosted on protocols.io. Read more information on sharing protocols at https://plos.org/protocols?utm_medium=editorial-email&utm_source=authorletters&utm_campaign=protocols.

We look forward to receiving your revised manuscript.

Kind regards,

Isabel Cristina Gonçalves Leite

Academic Editor

PLOS ONE

Journal Requirements:

**Additional Editor Comments:**

Your article requires, according to the reviewers, minor corrections, as described below.

Reviewers' comments:

Reviewer's Responses to Questions

**Comments to the Author**

1. Is the manuscript technically sound, and do the data support the conclusions?

Reviewer #1: Yes

Reviewer #2: Yes

2. Has the statistical analysis been performed appropriately and rigorously? 

Reviewer #1: Yes

Reviewer #2: Yes

3. Have the authors made all data underlying the findings in their manuscript fully available?

Reviewer #1: Yes

Reviewer #2: Yes

4. Is the manuscript presented in an intelligible fashion and written in standard English?

Reviewer #1: Yes

Reviewer #2: Yes

5. Review Comments to the Author

Reviewer #1: The article has a clear and well-structured objective. In the introduction, I recommend briefly conceptualizing Andersen's model. The discussion addresses limitations and strengths. Another study by the same authors has already been published in the journal with the title "Contextual and individual factors associated with public dental services utilization in Brazil: A multilevel analysis" presenting data from the same research in a different approach.

Reviewer #2: According to my understanding, constructing and validating a theoretical model of relationships between dental services use and socioeconomic characteristics, oral health status, primary care coverage, and public dental services was the aim of this study for further investigations, providing evidence of a public access policy’s effect on oral health services on equity and supporting the construction of more effective and equitable public policies. Fortunately, This study successfully defined a theoretical model. Socioeconomic status was negatively associated with oral health status, enrollment in primary care facilities, and the use of public dental consultations. Being black, indigenous, or living in a rural area was directly associated with lower socioeconomic status and greater use of public dental services.

It is an interesting study!

I am satisfied with this article in my initial review. I recommend that the article be accepted.

6. PLOS authors have the option to publish the peer review history of their article (what does this mean?). If published, this will include your full peer review and any attached files.

Reviewer #1: No

Reviewer #2: No

---

## [Author Response · Author response to Decision Letter 0]

16 Aug 2023

We are pleased to confirm that all the points raised by the academic editor and reviewers have been duly addressed and incorporated into the revised version of our manuscript.

1-Conceptualization of the Andersen Model:

We acknowledge the suggestion made by Reviewer 1 to include a conceptualization of the Andersen model in the introduction. We have revised the introduction to provide an overview of the Andersen Model. This addition enhances the context and helps readers better understand the theoretical framework of the theme.

2-Objective Refinement:

We thank Reviewer 2 for the insightful comments regarding refining our research objectives. In response to their suggestions, we have rephrased the objectives. These revised objectives enhance the precision of our research goals, contributing to a more focused and coherent presentation.

3-Reference List Review:

Furthermore, we would like to highlight that we have thoroughly reviewed our reference list to ensure its completeness and accuracy. Alterations have been made as necessary to ensure the citations' accuracy.

---

## [Editor Report · Decision Letter 1]

21 Aug 2023

Explaining public dental service utilization: a theoretical model

PONE-D-23-19396R1

Dear Dr. Galvão,

We’re pleased to inform you that your manuscript has been judged scientifically suitable for publication and will be formally accepted for publication once it meets all outstanding technical requirements.

Kind regards,

Isabel Cristina Gonçalves Leite

Academic Editor

PLOS ONE

---

## [Editor Report · Acceptance letter]

23 Aug 2023

PONE-D-23-19396R1 

Explaining public dental service utilization: a theoretical model 

Dear Dr. Galvao:

I'm pleased to inform you that your manuscript has been deemed suitable for publication in PLOS ONE. Congratulations! Your manuscript is now with our production department. 

Kind regards, 

on behalf of

Dr. Isabel Cristina Gonçalves Leite 

Academic Editor

PLOS ONE